# Preterm Infant Outcomes Following COVID-19 Lockdowns in Melbourne, Australia

**DOI:** 10.3390/children8121169

**Published:** 2021-12-10

**Authors:** Brendan Mulcahy, Daniel L. Rolnik, Alexia Matheson, Yizhen Liu, Kirsten R. Palmer, Ben W. Mol, Atul Malhotra

**Affiliations:** 1Monash Newborn, Monash Children’s Hospital, Melbourne, VIC 3168, Australia; mulcahy.brendan@gmail.com; 2Department of Obstetrics and Gynaecology, Monash University, Melbourne, VIC 3168, Australia; daniel.rolnik@monash.edu (D.L.R.); ayzliu@gmail.com (Y.L.); kirsten.palmer@monash.edu (K.R.P.); 3Department of Obstetrics and Gynaecology, Monash Health, Melbourne, VIC 3168, Australia; alexiawmatheson@gmail.com (A.M.); ben.mol@monash.edu (B.W.M.); 4Aberdeen Centre for Women’s Health Research, School of Medicine, University of Aberdeen, Aberdeen AB24 3FX, UK; 5Department of Paediatrics, Monash University, Melbourne, VIC 3168, Australia

**Keywords:** coronavirus, inflammatory markers, morbidity, mortality, newborn

## Abstract

Background Community lockdowns during the coronavirus disease 2019 (COVID-19) pandemic may influence preterm birth rates, but mechanisms are unclear. Methods We compared neonatal outcomes of preterm infants born to mothers exposed to community lockdowns in 2020 (exposed group) to those born in 2019 (control group). Main outcome studied was composite of significant neonatal morbidity or death. Results Median gestational age was 35 + 4 weeks (295 infants, exposed group) vs. 35 + 0 weeks (347 infants, control group) (*p* = 0.108). The main outcome occurred in 36/295 (12.2%) infants in exposed group vs. 46/347 (13.3%) in control group (*p* = 0.69). Continuous positive airway pressure (CPAP) use, jaundice requiring phototherapy, hypoglycaemia requiring treatment, early neonatal white cell and neutrophil counts were significantly reduced in the exposed group. Conclusions COVID-19 community lockdowns did not alter composite neonatal outcomes in preterm infants, but reduced rates of some common outcomes as well as early neonatal inflammatory markers.

## 1. Introduction

The coronavirus disease 2019 (COVID-19) pandemic has had a far-reaching impact on maternal and infant wellbeing [1,2]. While COVID-19 infection in pregnancy has been associated with an increased risk of preterm birth [3,4], several studies have suggested a reduction in preterm births in non-infected mothers during community lockdown [5,6,7,8,9]. Very few studies have assessed what impact this has had on neonatal outcomes [10,11], and previous studies are from countries with relatively high COVID-19 community transmission rates.

Melbourne, Australia, experienced one of the world’s strictest community transmission mitigation measures during two lockdowns (March–May, July–November) in 2020. These measures coupled with relatively low rates of community transmission then make us uniquely placed to examine the effects of lockdown measures on neonatal outcomes. Furthermore, some studies have suggested reduced exposure to infective pathogens as a reason for reduced preterm birth during lockdown [5,6,7,8]. We aimed to explore effects of community lockdowns on neonatal outcomes of preterm infants, with a secondary focus on the underlying rates of systemic inflammation in infants and mothers around the time of birth.

## 2. Methods

We conducted a retrospective cohort study at Monash Health, Melbourne’s largest public health service (>10,000 births per year). Data were gathered from the network’s birth registry and from electronic medical records. Ethical approval was obtained from Monash Health Human Research Ethics Committee (approval number QA/69113/MonH-2020-235157). Preterm infants (>20 weeks and <37 weeks’ gestation) born to mothers who conceived between 1 November 2019 and 29 February 2020, who were between 3 and 19 weeks pregnant at the beginning of lockdown (exposed group), compared to preterm infants conceived at the same time period the year before (control group) were included [3]. Conception was calculated based on gestational age from first day of the last menstrual period and confirmed with first trimester ultrasound.

We compared neonatal outcomes of preterm infants born during lockdown in 2020 (exposed group) with preterm infants born in 2019 (control group). The main outcome studied was a composite of death or significant morbidity in preterm infants, including chronic lung disease (CLD), severe (grade III-IV) intraventricular haemorrhage (IVH), necrotising enterocolitis (NEC), retinopathy of prematurity (ROP) requiring treatment, or culture positive sepsis. The secondary outcomes were rates of individual preterm complications and differences in inflammatory markers between cohorts.

For comparison of inflammatory markers we chose infants born <34 weeks, along with their mothers, due to local hospital protocol to screen these infants for infection. Included were first neonatal C-reactive protein (CRP) and full blood count (FBC) within seven days of birth, and last maternal CRP and FBC prior to delivery. Neonatal total white cell count (WCC), neutrophil count and immature-to-total neutrophil (IT) ratio were recorded. Categorical variables were compared using chi-squared or Fisher’s Exact test. Continuous variables were evaluated for normality of distribution. Non-normally distributed continuous variables were compared using the Wilcoxon rank-sum test and normally distributed variables using independent-samples t-tests. Effect estimates are reported as odds ratios (OR) of exposed compared to controls with 95% confidence intervals (CI). *p* values below 0.05 were considered statistically significant. Analyses were performed using Stata statistical package (Release 16, College Station, TX, USA).

## 3. Results

There were 295/3187 (9.3%) live neonates born <37 weeks in the exposed group, versus 347/3229 (10.7%) live neonates in the control group (OR 0.85; 95% CI 0.72–1.00; *p* = 0.047). Median gestational age was 35 + 4 weeks in the exposed group (IQR 33 + 1 to 35 + 3 weeks) compared with 35 + 0 weeks (32 + 3 to 36 + 2 weeks) in the control group (*p* = 0.108).

The primary outcome occurred in 36 (12.2%) infants <37 weeks in exposed group, compared with 46 (13.3%) infants <37 weeks in control group (OR 0.91; 95% CI 0.57–1.45; *p* = 0.69). When the mother was considered the unit of analysis, the primary outcome due to prematurity occurred in 36/3187 (1.1%) versus 46/3229 (1.4%) of live neonates born before 37 weeks in the exposed group and non-exposed group (RR 0.79, 95% CI 0.51 to 1.20).

Secondary outcomes showed a significant reduction in continuous positive airway pressure (CPAP) requirement, jaundice requiring phototherapy, hypoglycaemia requiring treatment, admission to the neonatal intensive care unit (NICU) > 48 h and median NICU length of stay in the exposed group (Table 1). There were no significant differences in other individual outcomes between the groups, including neonatal death.

Significant reductions in neonatal white cell counts (9.2, IQR 6.2–12.0 vs. 10.7, IQR 7.9–13.8; *p* = 0.01) and neutrophil counts (3.1, IQR 1.8–4.9 vs. 3.8, IQR 2.3–5.7; *p* = 0.04) were seen in the exposed group compared to controls. There were no significant differences in maternal inflammatory markers (Table 2).

## 4. Discussion

While we found a 15% reduction in preterm births <37 weeks gestation in the lockdown cohort, there was no significant difference in the composite primary neonatal outcome among preterm infants. Seasonal variability in preterm births was not assessed, given both cohorts encompassed the same time period in adjacent calendar years.

Among preterm infants, there was a 33% reduction in the odds of needing NICU admission, a 41% reduction in CPAP requirement, a 34% reduction in neonatal jaundice requiring phototherapy and a 45% reduction in hypoglycaemia requiring intravenous therapy in the exposed group. The reduction in these common neonatal complications could reflect the shift seen towards a more mature gestational age and significantly higher birth weight (mean 125 g heavier) of infants in the exposed group as compared to unexposed group. Ours is the first study that we are aware of to report on as comprehensive a range of neonatal outcomes following lockdown measures, particularly in light of the reduction in premature births observed previously [6]. There are limited studies that reported on neonatal outcomes following lockdown [11,12]. They did not show any statistically significant reductions in individual neonatal outcomes. The authors of a systematic review of perinatal outcomes during the pandemic era were only able to report comparable data for three outcome measures (five-minute Apgar scores <7, birth weight <2500 g and NICU admission) to include in their meta-analysis, due to a lack of studies assessing any other specific neonatal outcomes [13]. Our results are almost the exact opposite of the findings from studies of infants born to mothers acutely affected by COVID-19, in whom an increase in NICU admission, respiratory distress and jaundice has been observed [10], reinforcing that our observations are likely due to the impact of community lockdown alone in a maternal population without high community COVID-19 infection rates.

This study is the first published evidence we are aware of to link lockdown to a possible reduction in inflammatory markers in the neonatal period. We found a 14% and 18% reduction in neonatal total white cell counts and neutrophil counts, respectively, for preterm infants born of mothers exposed to lockdown. It is plausible that the reduction of preterm birth rates in lockdown is at least in part due to reduced infant exposure to inflammation processes, as had been hypothesised previously [5,6,7,8]. A reduction in other common viral infections has indeed been noticed during COVID-19 community lockdowns, which may be contributory to decreased inflammation observed in the perinatal period [14,15,16]. Future studies could consider assessment of other inflammatory markers such as the NLRP3 inflammasome that have been identified as having a role in the pathogenesis of clinical disease in COVID-19 infection [17]. Furthermore, the role of altered placental inflammation may be important to explore in the future.

Strengths of this study include it being the first to assess such a broad range of neonatal outcomes in relation to lockdown measures. Limitations include the retrospective design and the relatively small numbers of preterm infants affected by significant morbidity, limiting the statistical power. Future studies should also explore the role of maternal stress on preterm birth and outcomes in populations exposed to community lockdowns.

## 5. Conclusions

Within Melbourne’s stringent COVID-19 lockdowns and low burden of community COVID-19 spread, we found no difference in significant morbidity or death between groups, but saw a reduction in preterm births <37 weeks, CPAP requirement, jaundice and hypoglycaemia among preterm infants born to mothers exposed to community lockdowns. A reduction in neonatal white cell counts and neutrophil counts was also seen in the exposed cohort. Further investigation is needed to confirm these findings and how they may be used to direct future lifestyle interventions that may reduce inflammation in pregnancy and help reduce prematurity rates.

## Figures and Tables

**Table 1 children-08-01169-t001:** Comparison of neonatal outcomes (live born neonates <37 weeks gestation).

Outcomes	Exposed Group (n = 295)	Unexposed Group (n = 347)	Odds Ratio or Mean Difference (95% CI)	*p* Value
**Gestational Age**, *median (IQR)*	35^+4^ (33^+1^–35^+3^)	35^+0^ (32^+3^–36^+2^)	-	0.108
**Birth weight (g)**, *mean (SD)*	2219 (763)	2094 (736)	125 (8–242)	0.004
**Primary composite outcome ^$^**	36 (12.2)	46 (13.3)	0.91 (0.57–1.45)	0.690
Chronic lung disease	15 (5.1)	25 (7.2)	0.69 (0.36–1.33)	0.268
IVH grade III or IV	7 (2.7)	3 (0.9)	2.79 (0.71–10.88)	0.124
Necrotising enterocolitis	2 (0.7)	7 (2.0)	0.33 (0.07–1.61)	0.150
Retinopathy of prematurity	4 (1.4)	2 (0.6)	2.37 (0.43–13.0)	0.306
Culture-positive sepsis	8 (2.7)	15 (4.3)	0.62 (0.26–1.48)	0.274
RDS requiring intubation	25 (8.5)	28 (8.1)	1.05 (0.60–1.85)	0.850
CPAP requirement	66 (22.4)	114 (32.9)	0.59 (0.41–0.84)	0.003
Jaundice requiring phototherapy	110 (37.3)	165 (47.6)	0.66 (0.48–0.90)	0.009
Hypoglycaemia requiring IVT	59 (20.0)	108 (31.1)	0.55 (0.38–0.80)	0.001
Osteopenia requiring treatment	17 (5.8)	33 (9.5)	0.58 (0.32–1.07)	0.077
PDA requiring treatment	9 (3.1)	14 (4.0)	0.75 (0.32–1.76)	0.504
Neonatal seizures	2 (0.7)	2 (0.6)	1.18 (0.16–8.41)	0.871
Hyponatraemia	18 (6.1)	26 (7.5)	0.80 (0.43–1.49)	0.487
Hypernatraemia	9 (3.1)	16 (4.6)	0.65 (0.28–1.50)	0.309
Admission to NICU >48 h	82 (27.8)	127 (36.6)	0.67 (0.48–0.93)	0.018
Median NICU stay, *days (IQR)*	5 (0–21)	9 (0–27)	-	0.019
Neonatal death	11 (3.7)	8 (2.3)	1.64 (0.65–4.14)	0.289

Data presented as a total number (percentage) unless specified. CI: confidence interval; CPAP: Continuous positive airway pressure; GA: gestational age; IQR: Interquartile range; IVH: intraventricular haemorrhage; IVT: intravenous therapy; LBW: low birth weight; NICU: Neonatal Intensive Care Unit; PDA: patent ductus arteriosus; RDS: respiratory distress syndrome; VLBW: very low birth weight. ^$^ Primary composite outcome = death or significant neonatal morbidity (chronic lung disease, severe IVH grade III or IV, necrotising enterocolitis, retinopathy of prematurity requiring treatment, culture positive sepsis, RDS requiring intubation).

**Table 2 children-08-01169-t002:** Comparison of inflammatory markers among preterm neonates and mothers who gave birth to a preterm neonate (<34 weeks gestation).

Inflammatory Markers	Exposed Group (n = 90)	Unexposed Group (n = 129)	*p*-Value
**Maternal results**			
C-reactive protein, in mg/L	12.0 (6.0–41.0)	17.0 (7.0–27.0)	0.924
White cell count, in 10^9^/L	13.5 (10.8–16.0)	13.8 (12.0–17.2)	0.193
Neutrophil count, in 10^9^/L	10.4 (8.2–12.9)	10.5 (8.7–13.9)	0.302
Blood culture + ve sepsis, *n* (%) *	1/74 (1.4)	1/106 (0.9)	0.797
**Neonatal results**			
C-reactive protein, in mg/L	0.2 (0–0.8)	0.2 (0–0.7)	0.991
White cell count, in 10^9^/L	9.2 (6.2–12.0)	10.7 (7.9–13.8)	0.011
Neutrophil count, in 10^9^/L	3.1 (1.8–4.9)	3.8 (2.3–5.7)	0.041
Immature cells, in 10^9^/L	0.3 (0.1–0.7)	0.2 (0.1–0.7)	0.478
Immature-to-total neutrophil ratio	0.10 (0.05–0.17)	0.06 (0.02–0.14)	0.019 **
Blood culture + ve sepsis, *n* (%)	4 (4.4)	6 (4.7)	0.943

Data presented as median (IQR 25th–75th percentile). IQR: interquartile range. * Denominator is number of pregnant women who delivered at least one live infant < 34 weeks. ** Although significantly different between groups, both ranges are within normal limits.

## Data Availability

Data available on reasonable request.

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
