# Peer review of "Preterm Infant Outcomes Following COVID-19 Lockdowns in Melbourne, Australia"

_children, 2021, doi:10.3390/children8121169_

Round 1
Reviewer 1 Report
I congratulate the authors for this excellent study. The authors found no difference in significant morbidity or death between 133 groups within Melbourne’s stringent COVID-19 lockdown and low burden of community COVID-19 spread but saw a reduction in preterm births < 37 weeks, CPAP requirement, jaundice, and hypoglycemia among preterm infants born to mothers exposed to community lockdowns. The seasonal variability seems to be missed. Moreover, the reduced rate of comorbidity should be part of a better discussion reporting other studies. Moreover, the placenta data have not been discussed. This data should be added, please. Future studies should involve the inflammasome markers and may need a small paragraph in the discussion (Sergi CM, Chiu B. Targeting NLRP3 inflammasome in an animal model for Coronavirus Disease 2019 (COVID-19) caused by the Severe Acute Respiratory Syndrome Coronavirus 2 (SARS-CoV-2). J Med Virol. 2021 Feb;93(2):669-670. doi: 10.1002/jmv.26461. Epub 2020 Sep 29. PMID: 32841451; PMCID: PMC7461176.).
Author Response
Answer: We thank the Reviewer for their encouraging comments and for taking the time to read our manuscript. We had not previously commented on seasonal variability given that both cohorts represented the same seasonal time periods in adjacent years. A comment has now been included in the manuscript to highlight this. We agree with the reviewer that further discussion on reduced neonatal comorbidities in relation to other studies would be of benefit to the manuscript and this has now been included within the discussion. We are unsure whether the reviewer’s suggestion of further placental data was in reference to placental swabs in the context of chorioamnionitis and possible infection, or more general placental outcomes, but have added a comment in discussion regarding this. We thank the reviewer for the interesting suggestion around the direction of future studies and have included this now within the discussion along with the reference suggested.
Reviewer 2 Report
This is a very interesting submission regarding the effect of covid-19 lockdown on preterm infant outcomes in Melbourne. Covid-19 has been a special situation and as such it is significant to have a study on this issue.
The study is original and presents data well.
Besides reduced infant exposure to inflamation processes and/or viral infections, author could discuss whether maternal stress under these conditions (reduced ?) could have an impact on these outcomes.
Minor comment: Delete 'c' in Abstract (2nd line)
Author Response
This is a very interesting submission regarding the effect of covid-19 lockdown on preterm infant outcomes in Melbourne. Covid-19 has been a special situation and as such it is significant to have a study on this issue.
The study is original and presents data well.
Besides reduced infant exposure to inflamation processes and/or viral infections, author could discuss whether maternal stress under these conditions (reduced ?) could have an impact on these outcomes.
Answer: Thank you the suggestion. We have added a comment regarding this in the discussion.
Minor comment: Delete 'c' in Abstract (2nd line)
Answer: We thank the reviewer for their kind comments and time taken in providing their feedback. We have amended the error in the abstract that should in relation to the minor comment that should have read “Methods”, thank you for bringing this to our attention.